

# Quality of life of older Chinese adults receiving primary care in Wuhan, China: a multi-center study

Bao-Liang Zhong[1,2], Yan-Min Xu[2], Wu-Xiang Xie[3] and Xiu-Jun Liu[2]

[1] Research Center for Psychological and Health Sciences, China University of Geosciences, Wuhan, Hubei Province, China
[2] Affiliated Wuhan Mental Health Center, Tongji Medical College of Huazhong University of Science & Technology, Wuhan, Hubei Province, China
[3] Peking University Clinical Research Institute, Peking University Health Science Center, Beijing, China

## ABSTRACT

**Background**. Quality of life (QOL) is an important primary care outcome, but the QOL of older adults treated in primary care is understudied in China. This study examined QOL and its associated factors in older adults treated in Chinese primary care.

**Methods**. A total of 752 older patients (65+ years) were consecutively recruited from 13 primary care centers in Wuhan, China, and interviewed with a standardized questionnaire, concerning socio-demographics, major medical conditions, loneliness, and depression. QOL and depression were measured with the Chinese six-item QOL questionnaire and the shortened Geriatric Depression Scale, respectively. Multiple linear regression was used to identify factors associated with poor QOL.

**Results**. The average QOL score of primary care older adults was (20.7 ± 2.5), significantly lower than that of the Chinese general population. Factors significantly associated with poor QOL of Chinese primary care older adults included engaging in manual labor before older adulthood (unstandardized coefficient [β]: −0.702, $P < 0.001$), no living adult children (β: −1.720, $P = 0.001$), physical inactivity (β: −0.696, $P < 0.001$), having ≥ four major medical conditions (β: −1.813, $P < 0.001$), hearing problem (β: −1.004, $P = 0.017$), depression (β: −1.153, $P < 0.001$), and loneliness (β: −1.396, $P < 0.001$).

**Conclusions**. Older adults treated in Chinese primary care have poorer QOL than the general population. Addressing psychosocial problems at Chinese primary care settings could be helpful in improving QOL in Chinese older adults.

Corresponding author
Xiu-Jun Liu,
yulongguowang@126.com

## INTRODUCTION

In China, the unprecedented social changes during the past four decades have posed significant challenges to the health and well-being of older adults: decreased family size, weakened traditional family cohesion, changes in living arrangements, rapid economic growth, fast urbanization and industrialization, and massive rural-to-urban migration (*Yu et al., 2016*; *Zhong et al., 2018a*; *Zhong et al., 2018b*). China is ageing much faster than almost any other country in the world in recent history, and, until now, it has been the

nation with the largest number of older adult population. In 2017, the total number of Chinese older adults (65+) had reached 158.31 million, over one-tenth of China's total population, and by 2050, this number will reach 336 million, nearly one-third of the total population (*National Bureau of Statistics of China, 2018*; *Zhong, Chiu & Conwell, 2016a*). However, the infrastructure of China has not been prepared to meet older adults' growing needs for healthcare and social services.

To solve the contradiction between increasing demands for healthcare services and limited medical services resources, China's healthcare reform since 1994 has focused on strengthening its primary health-care system and made substantial progress (*Li et al., 2017*). For example, in rural regions of China, where there are insufficient healthcare resources, in 1994, 11.2% of the villages had no any medical facilities and the average number of village doctors was 0.90 per village, while by 2017, 100% of the villages had primary care clinics and the average number of village doctors had increased to 1.6 per village, respectively (*Center for Statistics and Information of Ministry of Health of the People's Republic of China, 1995*; *China National Health and Family Planning Commission, 2018*). In China, because distance and transportation from home to health facility are two major determinants of older adults' preferred choice of health facility for care, 53.6% urban older adults seek treatment at community primary care centers unless they are seriously ill (*Han & Jin, 2016*). Due to the inconvenient transportation in rural regions, Chinese older adults living in these rural areas are more likely to seek treatment from local primary care clinics at their villages. Therefore, primary care has been very well-placed to provide general healthcare services for Chinese older adults (*Li et al., 2016*; *Li et al., 2017*; *Zhong et al., 2018b*).

Although the bio-psychosocial model has dominated medical practice for many years, medical services provided by primary care in contemporary China are still quite basic and largely limited to disease treatment (*Liang, Mays & Hwang, 2018*). Late life is a period when increasing numbers of psychosocial problems such as cognitive decline, depression, and loneliness are more likely to occur (*Zhong et al., 2017*). Due to unawareness of the importance of mental health services and insufficient capacity in managing psychosocial problems, Chinese older adults treated in primary care have greater unmet late-life needs for psychosocial services (*Sun, Lam & Wu, 2018*). Accordingly, the World Health Organization (WHO) advocated the integration of mental health services into primary healthcare, particularly in low- and middle-income countries such as China (*Ventevogel, 2014*).

Quality of life (QOL) is an important outcome measure of health-care practice, which is broader than the definition of health and defined as a sense of well-being that encompasses physical health, role functioning, social functioning, and psychological health (*Gu, Xu & Zhong, 2018*; *Post, 2014*; *Yu et al., 2016*). To overcome the limitation of current disease-centered treatment in Chinese primary care, it is necessary to include QOL as an important therapeutic target of primary health-care. Therefore, examining QOL and its predictors in older adults treated in primary care is the first necessary step towards health policy-making. However, although QOL of Chinese older adults has been extensively studied, most existing studies focused on QOL of community- and institution-dwelling older adults (*Chen, Hicks & While, 2013*; *Xiao, Yoon & Bowers, 2017*; *Zhu et al., 2018*), and, as far as we know, few

studies have investigated QOL of older adults who seek treatment at Chinese primary care settings. This study was set out to investigate QOL and its associated factors in older adults treated in Chinese primary care.

## MATERIALS & METHODS

### Participants

This was part of a large-scale cross-sectional multi-center study, which investigated a range of mental health outcomes, QOL, and loneliness among older primary care patients in Wuhan, China, the largest metropolitan city with over ten million residents in central-south China, from October 2015 to November 2016 (*Zhong et al., 2018b*). Wuhan is divided into 13 districts (seven urban and six rural), with populations ranging from 0.21 to 1.34 million. Considering the geographic representativeness of the study sample, we consciously selected one primary care center from each district, which was located near the center of the most populous area of the district. Older adults who were 65 years old or over and sought treatment at these primary care centers, were consecutively invited to participate in this study. We excluded older patients who were unable to complete the interview due to severe physical illnesses and severe cognitive impairment, as well as those with psychotic disorders.

The study was approved by the Institutional Review Board of Wuhan Mental Health Center (approval number: WMHC-IRB-S065). All participants provided informed consent prior to the interview.

### Procedures and instruments

This was a questionnaire survey. Before the main study, the questionnaire was pilot-tested and finalized. The questionnaire was distributed in a face-to-face interview manner. Interviewers were trained primary care physicians (PCPs) from the 13 primary care centers.

Demographic variables collected in the questionnaire included gender, age, education, marital status, main occupation before older adult hood (mental vs. manual labor), residence location (urban vs. rural), living arrangement (with family members, alone, with others), total number of living adult children, smoking behavior, and physical activity.

Currently smoking was defined as smoking at least one cigarette per day on at least five days per week (*Zhong et al., 2018b*). Subjects who regularly participated in physical exercise were defined as being physically active.

A checklist was used to collect data on patients' major medical conditions, which included 13 specific physical illnesses: hypertension, diabetes, heart disease, stroke and other cerebrovascular diseases, chronic obstructive pulmonary disease, cancer, tuberculosis, chronic prostatitis, chronic gastric ulcer, Parkinson's disease, anemia, hepatic sclerosis, and arthritis.

The presence of hearing and vision problems was operationally defined by the authors (*Zhong et al., 2018b*). A hearing problem was present if the interviewer must speak at a louder volume than usual to help the interviewee hear questions clearly, while a vision problem was present if the respondent reported having difficulties in seeing TV or movies.

Depressive symptoms were assessed with the validated Chinese shortened version of the Geriatric Depression Scale (GDS), which had 15 items and were all answered in a yes/no format (*D'Ath et al., 1994*; *He, Xiao & Zhang, 2008*; *Liu et al., 2013*). The total score of GDS ranged from zero to 15, with a cut-off score of five or more suggesting clinically significant depression.

In accordance with previous studies (*Dahlberg et al., 2015*; *Victor, Grenade & Boldy, 2005*; *Zhong et al., 2017*; *Zhong et al., 2018b*; *Zhong et al., 2018c*), feelings of loneliness were assessed with one single question: "How often do you feel lonely?". The question was responded on a five-point Likert scale: 1 = always, 2 = often, 3 = sometimes, 4 = seldom, 5 = never. Loneliness was present if the answer was "sometimes", "often", or "always".

The outcome of this study, QOL, was evaluated with the Chinese six-item QOL questionnaire (Table S1), which was developed by *Phillips et al. (2002)* and has been widely used to assess the QOL of various populations in China, including older adults (*Dong et al., 2013*; *Liu et al., 2013*; *Wu et al., 2017*; *Ye et al., 2013*; *Zhang et al., 2012*). The questionnaire had six questions and each assessed one domain of QOL (physical health, psychological health, economic circumstances, activities, family relationship, and relationships with non-family associates) on a five-point scale: 1 = very poor, 2 = poor, 3 = fair, 4 = good, 5 = very good. The total QOL score varied between six and 30, with higher score denoting better QOL. In this study, the internal consistency (Cronbach $\alpha$ coefficient) of this QOL questionnaire was 0.827.

## Statistical analysis

The average QOL score was calculated. One-sample $t$-test was used to compare QOL between primary care older adults and the normative data, which was derived from a cross-sectional survey with a very large representative sample ($n = 23,987$) of Chinese general adult population in 2004–05 (*Zhang et al., 2012*). We quantified the magnitude of the difference in QOL between older primary care patients and the general population with Hedges' g, a measure of standardized difference between two means. Hedges' g values of <0.50, 0.50–0.80, and >0.80 represent small, medium, and large differences, respectively (*Zhong et al., 2019*). QOL scores of different older adult cohorts according to demographic, clinical, and psychosocial characteristics were compared with independent-samples $t$-test or one-way analysis of variance (ANOVA) as appropriate. Multivariable linear regression analysis that entered all statistically significant variables in the above univariate analysis as independent variables and QOL score as the outcome variable was conducted to examine factors associated with QOL. Factors were selected with a backward stepwise method. Data analyses were conducted with SPSS version 17.0. The statistical significance level was set at $p < 0.05$ (two-sided).

## RESULTS

Altogether, 791 older adults treated in primary care were invited to join the study. Among them, ten were rejected, 15 were excluded due to severe cognitive impairment, six withdrew informed consent, and eight had missing values on variables of interest of the current analysis. A final sample of 752 older adults were included into the current analysis.

Mean age of the final sample was 73.0 years (standard deviation [SD]: 6.1, range: 65–97), and 53.9% were women. Other demographic, clinical, and psychosocial characteristics are displayed in Table 1.

The average QOL score of the whole sample was 20.7 (SD: 2.5), without significant gender difference (males vs. females: 20.7 [SD: 2.5] vs. 20.6 [SD: 2.5], $t = 0.178$, $P = 0.859$). Primary care older adults had statistically significant lower QOL score than the normative data of Chinese general population (20.7 vs. 23.0, $t = 25.475$, $P < 0.001$), which corresponded to a Hedges' g of 0.85, indicating a large magnitude of the difference in QOL between the two populations.

Results of univariate analysis (Table 1) show that QOL scores were significantly lower in older adults who were illiterate, had marital status other than married (never-married, separated, divorced, widowed, cohabitating, and remarried), engaged in physical labor before older adulthood, lived alone or with others, had no living adult children, were physically inactive, suffered from four or more major medical conditions, had hearing problem, were depressed, and felt lonely ($P \leq 0.001$).

In multiple linear regression analysis (Table 2), factors significantly associated with poor QOL were engaging in manual labor before older adulthood (unstandardized coefficient [β]: −0.702, $P < 0.001$), no living adult children (β: −1.720, $P = 0.001$), physical inactivity (β: −0.696, $P < 0.001$), having ≥four major medical conditions (β: −1.813, $P < 0.001$), hearing problem (β: −1.004, $P = 0.017$), depression (β: −1.153, $P < 0.001$), and loneliness (β: −1.396, $P < 0.001$).

## DISCUSSION

In recent decades, QOL has been increasingly emphasized as an important health care outcome, but it remains a neglected area for public policy of the Chinese primary health-care system. To the best of our knowledge, this is the first study investigating QOL of older adults treated in Chinese primary care. In our study, a statistically poorer QOL in Chinese primary care older adults as compared to the general population and the large difference in QOL between the two populations were found. Because these primary care older adults all had physical illnesses, a statistically and clinically poorer QOL is expected. Further, because aging is often related to a decreasing social network, reduced income and poor health (Yu et al., 2016), the subjective well-being of older adults is vulnerable to functional disabilities and psychosocial problems. As evident in our study, the prevalence of vision problem, depression, and loneliness in Chinese primary care older adults were as high as 10.1%, 30.6%, and 26.2%, respectively (Table 1).

This study identified a number of demographic, clinical, and psychosocial factors associated with decreased QOL in older adults treated in Chinese primary care. Since older women are more likely to experience functional impairment in mobility and psychosocial problems, QOL of older adults is generally lower in women than men (Hajian-Tilaki, Heidari & Hajian-Tilaki, 2017). However, we found similar levels of QOL between males and females, which is possibly due to the prevailing physical illnesses of the study sample-masking the effect of gender. Previous population-based studies have reported a significant
**Table 1  Characteristics of Chinese older adults treated in primary care and quality of life (QOL) scores by variable.**

| Characteristics | | Number of older adults | % | QOL score (mean ± standard deviation) | t/F | P |
|---|---|---|---|---|---|---|
| Gender | Male | 347 | 46.1 | 20.7 ± 2.5 | 0.178 | 0.859 |
| | Female | 405 | 53.9 | 20.6 ± 2.5 | | |
| Age (years) | 65–74 | 484 | 64.4 | 20.7 ± 2.5 | 1.263 | 0.207 |
| | 75 + | 268 | 35.6 | 20.5 ± 2.6 | | |
| Education | Illiterate | 177 | 23.5 | 20.1 ± 2.5 | 10.974 | <0.001 |
| | Primary school | 213 | 28.3 | 20.3 ± 2.4 | | |
| | Junior middle school | 212 | 28.2 | 20.9 ± 2.4 | | |
| | Senior middle school and above | 150 | 19.9 | 21.5 ± 2.6 | | |
| Marital status | Married | 534 | 71.0 | 20.9 ± 2.4 | 3.414 | 0.001 |
| | Others[a] | 218 | 29.0 | 20.2 ± 2.7 | | |
| Main occupation before older adulthood | Mental labor | 216 | 28.7 | 21.4 ± 2.5 | 5.461 | <0.001 |
| | Manual labor | 536 | 71.3 | 20.3 ± 2.5 | | |
| Residence place | Urban | 403 | 53.6 | 20.7 ± 2.6 | 0.905 | 0.366 |
| | Rural | 349 | 46.4 | 20.6 ± 2.4 | | |
| Living arrangement | With family members | 637 | 84.7 | 20.8 ± 2.5 | 15.229 | <0.001 |
| | Alone | 81 | 10.8 | 20.1 ± 2.6 | | |
| | With others | 34 | 4.5 | 19.2 ± 2.1 | | |
| Number of living adult children | 0 | 19 | 2.5 | 17.7 ± 2.2 | 5.854 | <0.001 |
| | ≥1 | 733 | 97.5 | 20.7 ± 2.5 | | |
| Currently smoking | No | 631 | 83.9 | 20.6 ± 2.5 | 0.283 | 0.777 |
| | Yes | 121 | 16.1 | 20.7 ± 2.5 | | |
| Physically active | No | 326 | 43.4 | 20.1 ± 2.6 | 5.447 | <0.001 |
| | Yes | 426 | 56.6 | 21.1 ± 2.4 | | |
| Number of major medical conditions | ≤3 | 680 | 90.4 | 20.9 ± 2.5 | 9.695 | <0.001 |
| | ≥4 | 72 | 9.6 | 18.7 ± 1.7 | | |
| Hearing problem | No | 722 | 96.0 | 20.7 ± 2.5 | 3.335 | 0.001 |
| | Yes | 30 | 4.0 | 19.2 ± 2.5 | | |
| Vision problem | No | 676 | 89.9 | 20.9 ± 2.5 | 1.050 | 0.294 |
| | Yes | 76 | 10.1 | 20.6 ± 2.5 | | |
| Depressive symptoms | No | 522 | 69.4 | 21.2 ± 2.3 | 8.898 | <0.001 |
| | Yes | 230 | 30.6 | 19.4 ± 2.6 | | |
| Loneliness | No | 555 | 73.8 | 21.2 ± 2.2 | 9.524 | <0.001 |
| | Yes | 197 | 26.2 | 19.1 ± 2.8 | | |

**Notes.**
[a]"Others" included never-married, separated, divorced, widowed, cohabitating, and remarried.

association between low socio-economic status and poor QOL (*Brennan et al., 2013*; *Lam et al., 2017*). In general, Chinese older adults who previously engaged in manual labor during their working age were often farmers and temporary workers of labor-intensive factories. Because contemporary China's social security system has not been well established, the majority of these older adults are only entitled to have a very low security level of basic endowment insurance and have to rely on family members for long-term care and health care (*Jiang, Yang & Sánchez-Barricarte, 2016*; *Zhong et al., 2018a*). The significant link

**Table 2  Multiple linear regression of factors significantly associated with poor quality of life.**

| Variable | Risk level | Reference level | Coefficient | Standard error | t | P |
|---|---|---|---|---|---|---|
| Main occupation before older adulthood | Manual labor | Mental labor | −0.702 | 0.176 | 3.986 | <0.001 |
| Number of living adult children | 0 | ≥1 | −1.720 | 0.510 | 3.372 | 0.001 |
| Physically active | No | Yes | −0.696 | 0.163 | 4.279 | <0.001 |
| Number of major medical conditions | ≥4 | ≤3 | −1.813 | 0.281 | 6.457 | <0.001 |
| Hearing problem | Yes | No | −1.004 | 0.421 | 2.384 | 0.017 |
| Depressive symptoms | Yes | No | −1.153 | 0.180 | 6.420 | <0.001 |
| Loneliness | Yes | No | −1.396 | 0.19 | 7.336 | <0.001 |

between manual labor before older adulthood and poor QOL in our study could be attributed to the low economic status of older adults who previously made their living by physical labor. In traditional Chinese culture, adult children are a major source of social support for older adults (*Zhong, Chiu & Conwell, 2016b*). Since social support plays an important role in buffering against the negative effects of stress, and protecting against physical and mental morbidities (*Gu, Xu & Zhong, 2018*), the relationship between no living adult children and poor QOL in our study might be explained by insufficient social support to older adults without adult children.

There is clear evidence that physical activity or exercise can reduce the risk of physical and mental health problems via multiple direct and indirect mechanisms, including lowering blood pressure, decreasing risk of sarcopenia, improve physical functions and body composition, improving sleep quality, providing opportunities for increased social contacts, and changing levels serotonin and endorphins in the brain (*Harris, 2018*; *Stubbs et al., 2018*; *Yoo et al., 2018*; *Warburton & Bredin, 2017*; *Westbury et al., 2018*). There is also supporting evidence for the beneficial effects of physical exercise on both physical and mental wellbeing (*Black et al., 2015*; *Trachte, Geyer & Sperlich, 2016*). Accordingly, we found significant association of physical inactivity with poor QOL of older adults treated in primary care.

Considering the negative effects of major medical conditions on physical QOL, and hearing problem on daily functioning, the significant associations of diminished QOL with more major medical conditions and hearing problem in our study are expected (*Gopinath et al., 2012*). The theory of QOL satisfaction model argues that unmet social needs are an important cause of reduced QOL (*Gu, Xu & Zhong, 2018*). Consistent with this theory, loneliness was found to be significantly associated with poor QOL in this study. Given the many deleterious effects of depression on both physical and mental health (*Zhong et al., 2015*), it is reasonable to find the significant association of depression with poor QOL. Overall, our findings on these physical and psychosocial correlates of poor QOL are consistent with previous studies (*Cao et al., 2016*; *Chen, Hicks & While, 2013*; *Gu, Xu & Zhong, 2018*; *Zhu et al., 2018*).

The present study has several limitations. First, this is a cross-sectional study, so the causality of associations between poor QOL and its associated factors could not be ascertained. Prospective studies are warranted to confirm these relationships. Second, due

to our limited research budget, no age- and gender-matched community-residing older adult controls were recruited. QOL comparison was made with the reported normative Chinese data. Third, we recruited older adults from primary care centers of only one city in China; primary care older adults of other cities were not included, particularly those from economically underdeveloped regions of China. We need to be cautious in generalizing our findings.

## CONCLUSIONS

In summary, older adults treated in Chinese primary care have poorer QOL than the general population in China. A variety of factors, particularly psychosocial problems, are significantly associated with poor QOL in Chinese primary care older adults. Given that psychosocial problems are preventable or modifiable, psychosocial services would be helpful for improving QOL of Chinese older adults in primary care settings. The significant associations of poor QOL with physical and psychosocial factors suggest that in addition to conventional disease treatment for older adults, it is necessary to integrate psychosocial services into Chinese primary health-care.

Although primary care is a promising setting for addressing psychosocial problems of Chinese older adults, there are still barriers to providing integrated mental health services in primary care settings in China. One of the most challenging barriers is PCPs' insufficient capacity for managing psychosocial problems in China (*Liang, Mays & Hwang, 2018*). Specialized educational and training programs are warranted to strengthen PCPs' ability to recognize, diagnose, manage and refer older patients with psychosocial problems and mental disorders. To improve older patients' physical and mental wellbeing, it is also necessary to train PCPs to acquire basic skills for psychosocial supports and health education. Considering the association between physical inactivity and poor QOL, PCPs should be aware of the negative effects of unhealthy lifestyle on the health and QOL of older adults and encourage their older patients to do appropriate physical activities such as Tai Chi and other traditional Chinese activities.

## ACKNOWLEDGEMENTS

The authors thank all the research staff for their team collaboration work and all the primary care physicians and older adults involved in this study for their cooperation and support.

### Funding

This work was supported by the National Natural Science Foundation of China (grant number: 71774060), the 2015 Irma and Paul Milstein Program for Senior Health Awards from the Milstein Medical Asian American Partnership Foundation, and the Wuhan Health and Family Planning Commission (grant number: WG16A02; WG14C24). The funders

had no role in study design, data collection and analysis, decision to publish, or preparation of the manuscript.

## Grant Disclosures

The following grant information was disclosed by the authors:

National Natural Science Foundation of China: 71774060.

2015 Irma and Paul Milstein Program for Senior Health Awards from the Milstein Medical Asian American Partnership Foundation.

Wuhan Health and Family Planning Commission: WG16A02, WG14C24.

## Competing Interests

Bao-Liang Zhong is an Academic Editor for PeerJ.

## Author Contributions

- Bao-Liang Zhong conceived and designed the experiments, analyzed the data, contributed reagents/materials/analysis tools, prepared figures and/or tables, authored or reviewed drafts of the paper, approved the final draft.
- Yan-Min Xu performed the experiments, contributed reagents/materials/analysis tools, approved the final draft.
- Wu-Xiang Xie analyzed the data, authored or reviewed drafts of the paper, approved the final draft.
- Xiu-Jun Liu conceived and designed the experiments, prepared figures and/or tables, approved the final draft.

## Human Ethics

The following information was supplied relating to ethical approvals (i.e., approving body and any reference numbers):

The study was approved by the Institutional Review Board of Wuhan Mental Health Center (approval number: WMHC-IRB-S065).

## Data Availability

Raw data is available as a Supplemental File.

## Supplemental Information

Supplemental information for this article can be found online at http://dx.doi.org/10.7717/peerj.6860#supplemental-information.

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
