# Peer review of "Quality of life of older Chinese adults receiving primary care in Wuhan, China: a multi-center study"

_PeerJ, doi:10.7717/peerj.6860_

## Round 0.1 · original submission · Minor Revisions

The two reviewers and I both see many positives from this manuscript, including the large sample size of Chinese older adults and use of multiple linear regression. There are however some issues identified by the two reviewers that need to be addressed for this can be more seriously considered for publication. Please be sure to attend to the reviewers comments as well as to the comments provided in the annotated manuscripts that they have submitted in their review.

Reviewer 1 ·

Basic reporting

The introduction provides background information and a rationale for the study. But please see my comments in the Edits section. Relevant literature is cited.

Experimental design

The study reports original research findings that fit the aims and scope of the journal.
A research question is stated, which is relevant and meaningful.
The investigation appears to have been conducted rigorously. Methods are described with sufficient information to be reproduced, but please see my comments in the Edits section.

Validity of the findings

The data is statistically sound.
The conclusions are appropriately stated and are connected to the original question investigated.

Additional comments

Edits
The following edits should be made to the manuscript and are listed based on where they appear in the manuscript.
1. The revised title should read: Quality of life of Chinese older adults treated in primary care in Wuhan, China: a multi-center study
2. Throughout the entire manuscript, please refer to older adults as older adults and not as OA. OA stands for osteoarthritis.
3. Throughout the manuscript replace no exercise habit with not physically active or physical inactivity (depending on the sentence structure).
4. Page one, line 33 should read no living adult children.
5. Page one lines 36 and 37 should read: Addressing psychosocial problems in Chinese primary care settings could be helpful in improving QOL in Chinese older adults.
6. On page 3 (the manuscript does not appeared to be numbered. But counting from the title page) please provide a clear sentence or a paragraph that links QOL to health status.
7. Page 3 line 97 should read: All participants provided informed consent prior to the interview.
8. Page 4 lines 108 and 109 should read: Participants who regularly participated in physical activity were defined as being physically active.

Annotated reviews are not available for download in order to protect the identity of reviewers who chose to remain anonymous.

·

Basic reporting

Some grammatical mistakes throughout the text and suggest to seek language editing service to improve on the English and expression. Some revision and comments are included in the annotated pdf file.

Experimental design

Research question defined, knowledge gap identified and statements were made to show how the study would fill the gap. Statement on ethical approval included. Informed consent obtained. Methods were described with sufficient information.

Validity of the findings

Statistical analysis is appropriate. Conclusions drawn were supported by the results, well stated and answer the original research question.

Additional comments

• Authors should standardize the use of terminology and use of English (British Vs American English)
• Discussion is sometimes a bit superficial, especially for paragraph line 201-212. Suggest more in-depth discussion by explaining the meaning of the findings and why they are important, and comparing the findings with other similar studies.
• Strength: Multiple linear regression analysis with reasonable sample size from multi-center, weaknesses were clearly noted by the authors
• Weakness: English language, meanings of some of the expressions are a bit vague and thoughts were not connected

---

## Round 0.2 · Minor Revisions

I thank the authors for attending to almost all of the reviewers’ comments on initial version of this manuscript. A few small things requiring amendments include the following specific points.

Specific comments

Please use the find and replace function to replace the word “elderly” with “older” or “older adult” in the small number of occasions that still occurs within the manuscript.
Line 36: you need to describe who the older adults treated in Chinese primary care have poorer quality of life than. Is it the general population or is it older adults not being treated in primary care?
Line 57: this should read “and village doctors has now risen to 1.14…”.
Line 183 – 184: you need to provide a reference year regarding the quality of life of the general population in China to support this comment. You should also go into more detail about the magnitude of these differences across the different domains of quality of life, both to the general population and to the wider older adult population who are not currently accessing primary care support if such data is available. Some more detail (perhaps a paragraph) about this would be most useful as it would highlight the areas of major significance that may require alterations to the healthcare system in China to minimise this discrepancy and quality of life.
Line 199: I am not sure what you mean by China social welfare system being ‘unsound”. Do you mean something like “is in its infancy compared to many Western countries” as not many older adults have savings in pensions that they can use in their old age?
Line 208 – 215: I would also suggest you add in here something about the effect of physical activity/exercise on improving body composition and physical function in older adults, particularly in light of the high prevalence and adverse effects related with sarcopenia in older populations.
Overall: could you provide some tentative recommendations about how the quality of life of older Chinese adults accessing primary care could be improved? For example, how may primary care professionals need to be provided with more resources or training to improve the quality of life of this segment of the population? Further, how may primary care sector work with other aspects of Chinese society to promote healthy behaviours to improve aspects of quality of life in these older patients? For example, could the promotion of a traditional Chinese activity such as Tai Chi be used effectively in this case?

Reviewer 1 ·

Basic reporting

The revisions are fine.

Experimental design

The revisions are fine.

Validity of the findings

The revisions are fine

Additional comments

Thank you for you revising your manuscript. I am satisfied with the revised manuscript.

---

## Round 0.3 · accepted · Accept

I thank the authors for their concerted efforts to take on board the comments of the reviewers and myself and am happy to recommend the manuscript be accepted for publication in PeerJ.

#